# Evolution of phenotypic plasticity leads to tumor heterogeneity with implications for therapy

**Simon Syga** [ID][1]*, **Harish P. Jain**[2], **Marcus Krellner**[3], **Haralampos Hatzikirou** [ID][1,4], **Andreas Deutsch**[1]

**1** Center for Interdisciplinary Digital Sciences, Department Information Services and High Performance Computing, TUD Dresden University of Technology, Dresden, Germany, **2** Njord Centre, Department of Physics, University of Oslo, Oslo, Norway, **3** School of Mathematics and Statistics, University of St Andrews, St Andrews, United Kingdom, **4** Mathematics Department, Khalifa University, Abu Dhabi, United Arab Emirates

\* simon.syga@tu-dresden.de

**Data Availability Statement:** The Python code used to generate the simulations for this study was published as a Zenodo snapshot here: doi.org/10. 5281/zenodo.10806326.

## Abstract

Cancer is a significant global health issue, with treatment challenges arising from intratumor heterogeneity. This heterogeneity stems mainly from somatic evolution, causing genetic diversity within the tumor, and phenotypic plasticity of tumor cells leading to reversible phenotypic changes. However, the interplay of both factors has not been rigorously investigated. Here, we examine the complex relationship between somatic evolution and phenotypic plasticity, explicitly focusing on the interplay between cell migration and proliferation. This type of phenotypic plasticity is essential in glioblastoma, the most aggressive form of brain tumor. We propose that somatic evolution alters the regulation of phenotypic plasticity in tumor cells, specifically the reaction to changes in the microenvironment. We study this hypothesis using a novel, spatially explicit model that tracks individual cells' phenotypic and genetic states. We assume cells change between migratory and proliferative states controlled by inherited and mutation-driven genotypes and the cells' microenvironment. We observe that cells at the tumor edge evolve to favor migration over proliferation and vice versa in the tumor bulk. Notably, different genetic configurations can result in this pattern of phenotypic heterogeneity. We analytically predict the outcome of the evolutionary process, showing that it depends on the tumor microenvironment. Synthetic tumors display varying levels of genetic and phenotypic heterogeneity, which we show are predictors of tumor recurrence time after treatment. Interestingly, higher phenotypic heterogeneity predicts poor treatment outcomes, unlike genetic heterogeneity. Our research offers a novel explanation for heterogeneous patterns of tumor recurrence in glioblastoma patients.

## Author summary

Intratumor heterogeneity presents a significant barrier to effective cancer therapy. This heterogeneity stems from the evolution of cancer cells and their capability for phenotypic

**Funding:** SS and AD are funded by the European Union (ERC, subLethal, 101054921, https://erc.europa.eu/). Views and opinions expressed are however those of the authors only and do not necessarily reflect those of the European Union or the European Research Council Executive Agency. Neither the European Union nor the granting authority can be held responsible for them. SS and AD acknowledge support by Worldwide Cancer Research (23-0177, https://www.worldwidecancerresearch.org). HPJ acknowledges support by the European Union's Horizon 2020 research and innovation programme (CompSci TraCS, 945371, https://cordis.europa.eu). HH thanks Volkswagenstiftung for its support in the "Life?" program (96732, https://www.volkswagenstiftung.de). He has received funding from the Bundesministerium für Bildung und Forschung under grant agreement No. 031L0237C (MiEDGE project/ERACOSYSMED, https://www.bmbf.de). He also acknowledges the support of the RIG-2023-051 grant from Khalifa University (https://www.ku.ac.ae) and the AJF-NIH-25-KU grant from the NIH-UAE collaborative call 2023 (https://www.aljalilafoundation.ae). The funders had no role in study design, data collection and analysis, decision to publish, or preparation of the manuscript.

**Competing interests:** The authors have declared that no competing interests exist.

plasticity. However, the interplay between these two factors still needs to be fully understood. This study examines the interaction between cancer cell evolution and phenotypic plasticity, focusing on the phenotypic switch between migration and proliferation. Such plasticity is particularly relevant to glioblastoma, the most aggressive form of brain tumor. By employing a novel model, we explore how tumor cell evolution, influenced by both genotype and microenvironment, contributes to tumor heterogeneity. We observe that cells at the tumor periphery tend to migrate, while those within the tumor are more inclined to proliferate. Interestingly, our analysis reveals that distinct genetic configurations of the tumor can lead to this observed pattern. Further, we delve into the implications for cancer treatment and discover that it is phenotypic, rather than genetic, heterogeneity that more accurately predicts tumor recurrence following therapy. Our findings offer insights into the significant variability observed in glioblastoma recurrence times post-treatment.

## Introduction

Cancer remains a significant challenge in public health despite improvements in clinical treatment. One of the leading causes of treatment failure is the emergence of therapy resistance, enabled by tumor heterogeneity [1]. Tumor heterogeneity can refer to the existence of genetically distinct tumor subclones or the existence of cancer cells with different phenotypic characteristics, including gene expression, metabolism, motility, proliferation, metastatic potential, and resistance to treatment [2, 3]. It results from genome instability and mutations, and non-genetic mechanisms of cancer progression and adaptation [4, 5]. Due to its heterogeneous nature, tumor growth is increasingly viewed as an evolutionary and ecological process in which abnormal cells compete for space and resources with each other and with healthy cells in the surrounding tissue [6, 7]. Lately, the ability of cancer cells to develop phenotypic plasticity has been proposed as a new hallmark of cancer [8]. Important examples include the epithelial to mesenchymal transition [9] and the change of metabolism from oxidative phosphorylation to anaerobic glycolytic metabolism (Warburg effect) [10]. Understanding phenotypic states and transitions, mechanisms driving plasticity, and how to measure and model these behaviors are significant challenges in the study of cancer plasticity [11].

One particularly interesting example of phenotypic plasticity is the migration-proliferation dichotomy, which has been observed for non-neoplastic cells [12] as well as during tumor development [13, 14]. This phenotypic switch is especially relevant in the context of glioblastoma, the most lethal form of brain cancer, with important implications for treatment [15]. The precise molecular mechanisms underlying this dichotomy remain poorly understood. The switch between migrating and proliferating phenotypes has been suggested to depend on the microenvironment of cells, such as growth factor gradients, extracellular matrix properties, or altered nutrient availability [9]. Because tumor cells secrete relevant factors, produce toxic metabolites, and consume oxygen and nutrients, their local cell density correlates with those chemicals' concentration [16, 17]. Therefore, the local cell density can be considered a proxy for analyzing the dependence of the switch on the tumor microenvironment.

Several mathematical models have shown that the migration-proliferation plasticity has a significant impact on tumor spread [18–22]. Böttger et al. [23] investigated the effect of a density-dependent switch between migrating and proliferating phenotypes on tumor persistence. They assumed that migration and proliferation are mutually exclusive and that cells switch between phenotypes according to a switching probability that depends on local cell density.

This approach led to the notion of attractive and repulsive go-or-grow strategies, meaning cells that tend to proliferate more or less with increasing cell density, respectively. They also showed that the attractive strategy could lead to a strong Allee effect, a situation in which the effective growth rate of a population can become negative for low cell densities [24]. Gallaher et al. [25] investigated the phenotypic evolution of cancer cells under proliferation-migration trade-offs. The authors assumed that the cellular phenotype is characterized by migration speed and proliferation rate and studied the effect of different trade-off conditions on the evolutionary dynamics. They showed that a higher migration rate is generally favored over proliferation at the tumor's edge and vice-versa in the tumor bulk. In a similar setting, the spatiotemporal phenotype distribution could be predicted by mean-field theory [26].

However, previous studies investigating the evolutionary dynamics of migration and proliferation traits did not explicitly consider the cell decision-making process governing the phenotypic switch between migration and proliferation. In these studies, all cells had an identical phenotypic switch regulation in response to microenvironmental changes, and the phenotypic selection was based solely on volume exclusion effects. This strongly simplified the biological complexity of cell decision-making. Cancer cells are phenotypically plastic and can adapt in response to their microenvironment according to their genotype, see Fig 1A. However, it is unknown how this phenotypic plasticity influences the evolutionary dynamics in general and in the context of the migration/proliferation switch in particular.

It remains to be determined whether there is an optimal regulation of the phenotypic switch between migration and proliferation and how the microenvironment might influence this. It is also uncertain why cells adopt an attractive go-or-grow strategy, particularly as this increases the risk of extinction due to the strong Allee effect [23]. Furthermore, the effects of go-or-grow plasticity on therapy outcomes have yet to be fully understood.

To tackle these questions, we propose a novel cellular automaton model for the emergence and subsequent natural selection of cells with different go-or-grow strategies. Cells switch between migratory and proliferative phenotypes as dictated by a phenotypic switch function that depends on their individual (fixed) genotype and the (variable) local microenvironment in the form of the local cell density relative to a cell density threshold (switch threshold), see Fig 1B. Consequently, the probability of either phenotype is not given by its genotype alone but by the combination of its genotype and microenvironment. The cells' genotypes can be

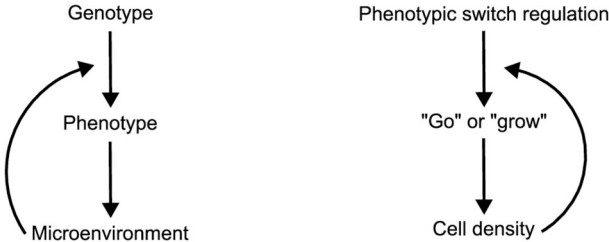

**A**
**Hierarchy of cell decision-making**

Genotype

Phenotype

Microenvironment

**B**
**Representation in mathematical model**

Phenotypic switch regulation

"Go" or "grow"

Cell density

**Fig 1. Hierarchical representation of cell decision-making.** (**A**) The cell's genotype dictates the regulation of the phenotypic switch. This switch determines the cell's reaction to its microenvironment. Subsequently, the cell interacts and shapes its microenvironment according to its phenotype. (**B**) In the mathematical model and the context of the go-or-grow dichotomy, the genotype is represented by the parameter $\kappa$ which controls the phenotypic switch between the migratory and proliferating phenotypes dependent on the local cell density. After assuming either phenotype the cell influences the local microenvironment by either reducing the cell density (by migration) or increasing it (by proliferation). We neglect possible epigenetic changes by a persistent microenvironment on long time scales.

inherited or acquired via mutations. We study the cellular automaton model by extensive computer simulations and a mathematical analysis based on a mean-field approximation. We perform a parameter scan varying the death rate and the cell density threshold while recording the evolutionary dynamics. We quantify the phenotypic and genetic heterogeneity that arises from these dynamics. Furthermore, our study extends to the implications of this heterogeneity for cancer therapy. We simulate cancer treatments and monitor the time until recurrence, which we correlate with the observed phenotypic and genetic tumor heterogeneity. Lastly, we discuss how medical evidence from glioblastoma patients supports the predictions made by our model.

## Methods

### Model definition

We define a stochastic, spatiotemporal, cell-based model to study the co-evolution of different go-or-grow strategies. This allows us to describe the cell decision-making of individual cells that each possess a unique go-or-grow strategy, which we associate with its genotype, see Fig 1B. We assume that a cell's decision-making depends on the local microenvironment. We use a cellular automaton model with a state space accounting for cell velocity (represented by channels). This class of cellular automata is called lattice-gas cellular automaton (LGCA) and originates in fluid dynamics simulations [27]. It was later extended to model biological phenomena such as excitable media, collective migration, and tumor growth [28–31].

In the LGCA, individual cells reside in channels on nodes of a regular lattice $\mathbf{r} \in \mathcal{L}$. Every node has $b$ nearest-neighbor nodes, where $b$ depends on the lattice geometry, e. g., for a one-dimensional lattice $b = 2$. Nodes are connected to their nearest neighbors by unit vectors $\mathbf{c}_i$, $i = 0, \ldots, b - 1$, which we call velocity channels. We have $c_0 = 1$, $c_1 = -1$ on a one-dimensional lattice. Each node has one rest channel (channels with zero velocity), $\mathbf{c}_b = 0$. The state of each node is updated independently and simultaneously in discrete time steps $k = 1, 2, 3, \ldots$. First, each state is updated in the interaction step using a stochastic update rule $\mathcal{R} : \mathbf{s} \to \mathbf{s}'$, incorporating all relevant biological mechanisms. Subsequently, cells residing in velocity channels are deterministically transported to nearest-neighbor nodes in the direction of the respective velocity channel, i. e.,

$$s_i(\mathbf{r} + \mathbf{c}_i, k + 1) = s_i'(\mathbf{r}, k). \tag{1}$$

The rules of our model are designed to implement the following assumptions about cellular behavior, see Fig 2.

- **Death.** All cells die with a constant probability $\delta$ in each time step.

- **Phenotypes.** Cells in the proliferating phenotype do not migrate, while cells with the migratory phenotype do not proliferate. Cells with the proliferative phenotype divide with fixed probability $\alpha(1 - n(\mathbf{r})/K)$, where $n(\mathbf{r})$ is the number of cells on the node and $K > 0$ is the carrying capacity, a free parameter.

- **Phenotypic switch.** Cells switch between a proliferating and a migratory phenotype depending on the local microenvironment and their unique phenotypic switch regulation $\kappa \in \mathbb{R}$. The probability for the proliferative phenotype is given by a function $r_\kappa(\rho_\mathcal{N}) \in (0, 1)$ that is determined by the cell's genotype $\kappa$ and depends on the local microenvironment $\rho_\mathcal{N}$. We define its exact form below.

- **Mutations.** During proliferation, the regulation of the phenotypic switch of the daughter cell $d$, $\kappa_d$, representing the cell's genotype, is drawn from a Gaussian distribution centered on the

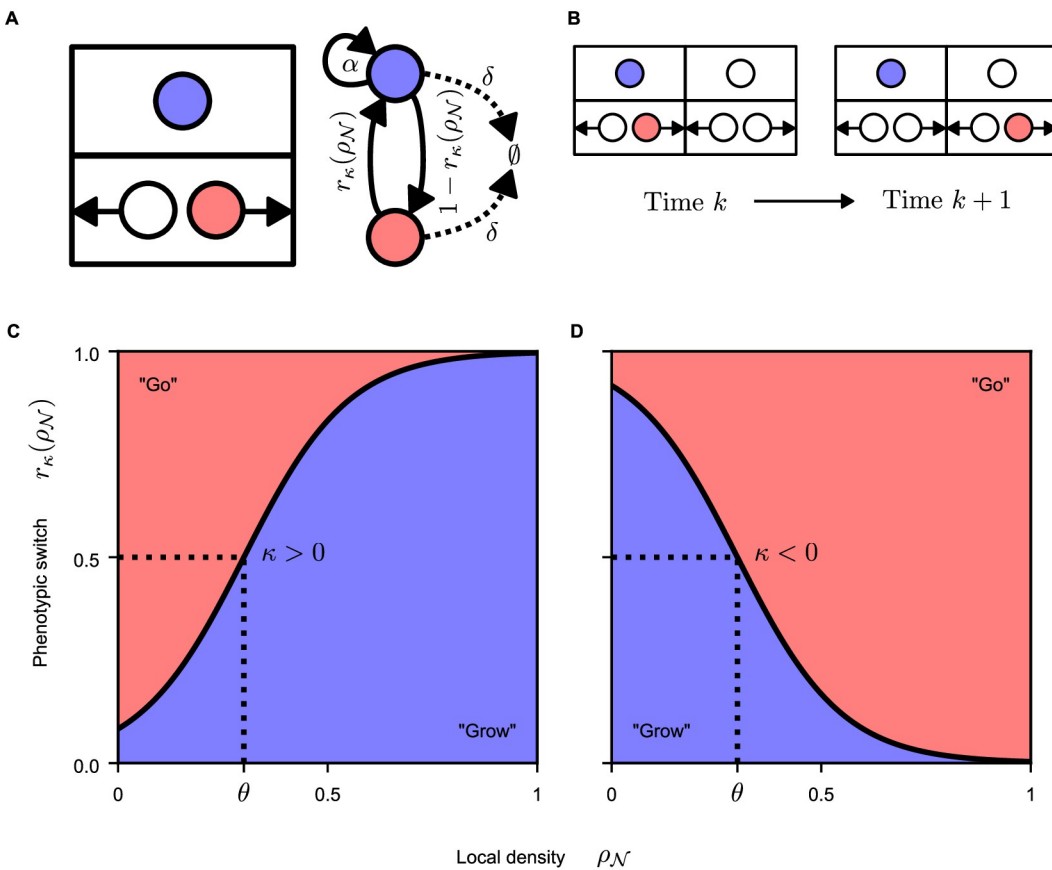

**Fig 2. The lattice-gas cellular automaton (LGCA) model.** (**A**) Left: The LGCA model is implemented on a one-dimensional lattice, where each node comprises two velocity channels for movement to the nearest-neighbor nodes and one rest channel for no movement. Each channel can be occupied by any number of cells. Cells in velocity channels (marked red) have the migratory phenotype, and cells in rest channels have the proliferative phenotype (blue). White channels denote the absence of tumor cells. Right: Schematic illustration of model dynamics: Cells switch between phenotypes depending on their genotype $\kappa$ and the local tumor cell density. Proliferative cells divide with a constant rate $\alpha$. Migratory cells randomly choose a new direction of movement. All cells die with a constant rate $\delta$. (**B**) Cell migration. Each migratory cell (red) moves in the direction of its respective velocity channel. (**C** and **D**) Phenotypic plasticity. The phenotypic switch probability $r_\kappa$ depends on the tumor cell density in the microenvironment $\rho_{\mathcal{N}}$ and the cell's genotype $\kappa$. The sign of $\kappa$ determines the switch regulation. (**C**) $\kappa > 0$ results in attractive behavior (proliferating phenotype triggered by high cell density), and $\kappa < 0$ (**D**) leads to repulsive behavior (cells switch to migratory phenotype if the local cell density becomes too high). The parameter $\theta$ indicates the cell density threshold, where the probability for either phenotype is 1/2.

respective parameter of the mother cell $\kappa_m$ and has a fixed standard deviation $\Delta\kappa$. This ensures that daughter cells inherit $\kappa$ values similar to those of the mother cells.

- **Migration.** Migratory cells perform an unbiased random walk.

Due to the variable proliferation behavior caused by the phenotypic switch between migration and proliferation, a constant cell death rate, and a steady influx of variation caused by the mutations, cells are subjected to Darwinian evolution, potentially leading to a complex spatial distribution of $\kappa$ values. This is the key novelty in this paper compared to [23], where all cells during one simulation followed the same phenotypic switch function $r_\kappa(\rho)$, i. e., had the same $\kappa$ value.

In the model, we divide the stochastic update rule $\mathcal{R}$ into several steps: First, a death operator removes dying cells. Next, all cells switch their phenotype according to the phenotypic

switch function, which depends on their regulation of the switch $\kappa$ and the average density in their microenvironment $\rho_{\mathcal{N}}(\mathbf{r}) \coloneqq \frac{1}{(b+1)K} \sum_{i=0}^{b} n(\mathbf{s}(\mathbf{r}+\mathbf{c}_i))$. Then, proliferative cells divide according to the probability mentioned above. Newly born cells are also placed in the rest channel, i. e., start out in the proliferative state. The switch parameter of the daughter cells is sampled from a Gaussian distribution centered on the switch parameters of the mother cells $\kappa_m$, i.e., $\kappa_l \sim \mathcal{N}(\kappa_m, \Delta\kappa^2)$. For a detailed definition of the mathematical model including a table of parameters, see S1 Text.

We propose that the transition between proliferative and migratory cellular behavior relies on the local cell density. We regard the cell density as a proxy for tumor cell interactions with extracellular matrix components, chemical signals, and stromal cells. These effects are modeled based on their correlation with cell density. Instead of reproducing the phenotypic switch process' intricacies involving signaling networks, we simplify the intracellular details into stochastic cell-based rules, yielding an analytically tractable model. This increases our understanding of the underlying dynamics.

Although the precise dependence of the phenotypic switch on cell density is uncertain, we opt for the simplest form: a monotonous dependence. This choice enables the discrimination of two complementary types of plasticity: attraction to or repulsion from densely populated regions. In the attraction scenario, cell motility decreases with local cell density, promoting proliferation in crowded areas, as shown in Fig 2C. Conversely, in the repulsion scenario, cells avoid densely populated areas by switching to the migratory phenotype and moving away while increasing proliferation in sparsely populated regions, as depicted in Fig 2D. An independent strategy is also possible where the phenotypic switch does not depend on cell density, i. e., $\kappa \approx 0$. Corresponding cells adopt either phenotype with equal probability irrespective of their microenvironment.

We assume that transitions between migrating and proliferating phenotypes, i. e., velocity and rest channels, are dictated by the phenotypic switch function

$$r_\kappa(\rho_{\mathcal{N}}) \coloneqq \frac{1}{2}\left(1 + \tanh(\kappa(\rho_{\mathcal{N}} - \theta))\right), \tag{2}$$

where $\rho_{\mathcal{N}}(\mathbf{r}) \coloneqq \frac{1}{(b+1)K} \sum_{i=0}^{b} n(\mathbf{r}+\mathbf{c}_i)$ represents the average density of tumor cells in the neighborhood, $\kappa$ is the unique regulation parameter of the switch and $\theta$ is a constant threshold parameter. The intensity of the dependence of the switch on the cell density is determined by the absolute value of $\kappa$. The sign of $\kappa$ indicates whether the cell applies the attractive ($\kappa > 0$) or the repulsive strategy ($\kappa < 0$) [23]. The cell density threshold at which the switching probabilities between phenotypes are equal is denoted by the parameter $\theta$. The functional form of the switching function was first suggested in a study that examined tumor invasion [20].

## Analysis of growth dynamics

We simulate the LGCA model on a one-dimensional lattice with $L = 1001$ nodes, corresponding to a maximum tumor diameter of 5 cm (see S1 Text), which is large enough that no cells reach the lattice boundary. We place $K$ cells in the central node's rest channel, i. e., these cells are in the proliferative phenotype, as an initial condition and choose their switch parameters $\kappa_l$ from a uniform distribution in the interval $[-4, 4]$. We investigate the impact of the death probability $\delta$ and the phenotypic switch threshold $\theta$ on evolutionary dynamics. To this end, we perform a parameter scan, systematically varying these parameters in the intervals $0 \leq \delta \leq 0.25$ and $0 \leq \theta \leq 1$. We fix the proliferation probability and capacity at $\alpha = 1$, $K = 100$ and the standard deviation $\Delta\kappa = 0.2$. For an overview of mathematical symbols and parameter values see Tables A and B in S1 Text.

We simulate the system for 1000 time steps, corresponding to about four years (see S1 Text), avoiding synthetic tumors reaching the lattice boundary at the end of the simulation. As observable, we record the spatial distributions of resting/migrating cells and the genotype distribution $\psi(r, \kappa)$.

We also perform simulations on two-dimensional hexagonal lattices to ensure that our results do not depend on the one-dimensional lattice. We choose a hexagonal lattice over a square lattice to reduce artificial patterns that can occur in spatially discrete models [32]. To this end, we place $K = 50$ cells in the middle of a hexagonal lattice of sidelength $L = 250$ and monitor the temporal dynamics for 300 time steps. Other parameters and observables are unchanged compared to the one-dimensional simulations. Due to the computational costs involved, we do not perform a systematic parameter scan but simulate the system for selected parameter sets to ensure that the observed dynamics in two dimensions match those in one dimension qualitatively.

### Characterization of therapy response

Starting from simulation endpoints of the growth dynamics simulations, we remove 99.9% of cells to simulate cancer therapy. This value was previously used to estimate the effect of the resection of a glioblastoma tumor [19]. We then let the tumor grow again until the number of cells has reached the level before treatment and record the required time as recurrence time.

### Quantification of heterogeneity

To quantify the heterogeneity in our synthetic tumors, we calculate the Shannon entropy of the observed distributions of phenotypes and genotypes. This information-theoretic score is maximal if every tumor cell subpopulation is equally probable and minimal if all cells are identical [33]. For the phenotype distribution, we can directly compute the entropy as

$$S_{\text{pheno}} := - p_{\text{migr}} \log_2 p_{\text{migr}} - p_{\text{prolif}} \log_2 p_{\text{prolif}}, \tag{3}$$

where $p_{\text{migr}}, p_{\text{prolif}}$ are the frequencies of migrating and proliferating cells, respectively. As the distribution of genotypes is continuous in our model, we apply a histogram-based estimator. We first bin our genotype values $\kappa_i$ using the method `histogram` of the Python package `numpy 1.24.2` with automatic bin estimation. Next, we can estimate the genotype entropy as

$$S_{\text{geno}} := - \sum_i n_i \log_2 \frac{n_i}{\Delta_i}, \tag{4}$$

where $n_i$ is the number of cells in bin $i$ and $\Delta_i$ is the width of bin $i$.

## Results

### Emergence of phenotypic and genetic heterogeneity

We first aim to characterize the emergent growth dynamics, starting with a small number of cells. Figs 3A–3F and S1–S3 show snapshots of representative examples of the observed evolutionary dynamics. S1–S6 Videos show representative growth dynamics in two dimensions. The population expands, invading the surrounding tissue reminiscent of a traveling wave. Interestingly, the spatial distribution of the switch parameter $\kappa$ shows a bimodal behavior for most combinations of parameters (see Fig 3E). In this case, a small subpopulation of cells with negative switch parameters $\kappa < 0$ leads to population growth at the edge, while most cells in

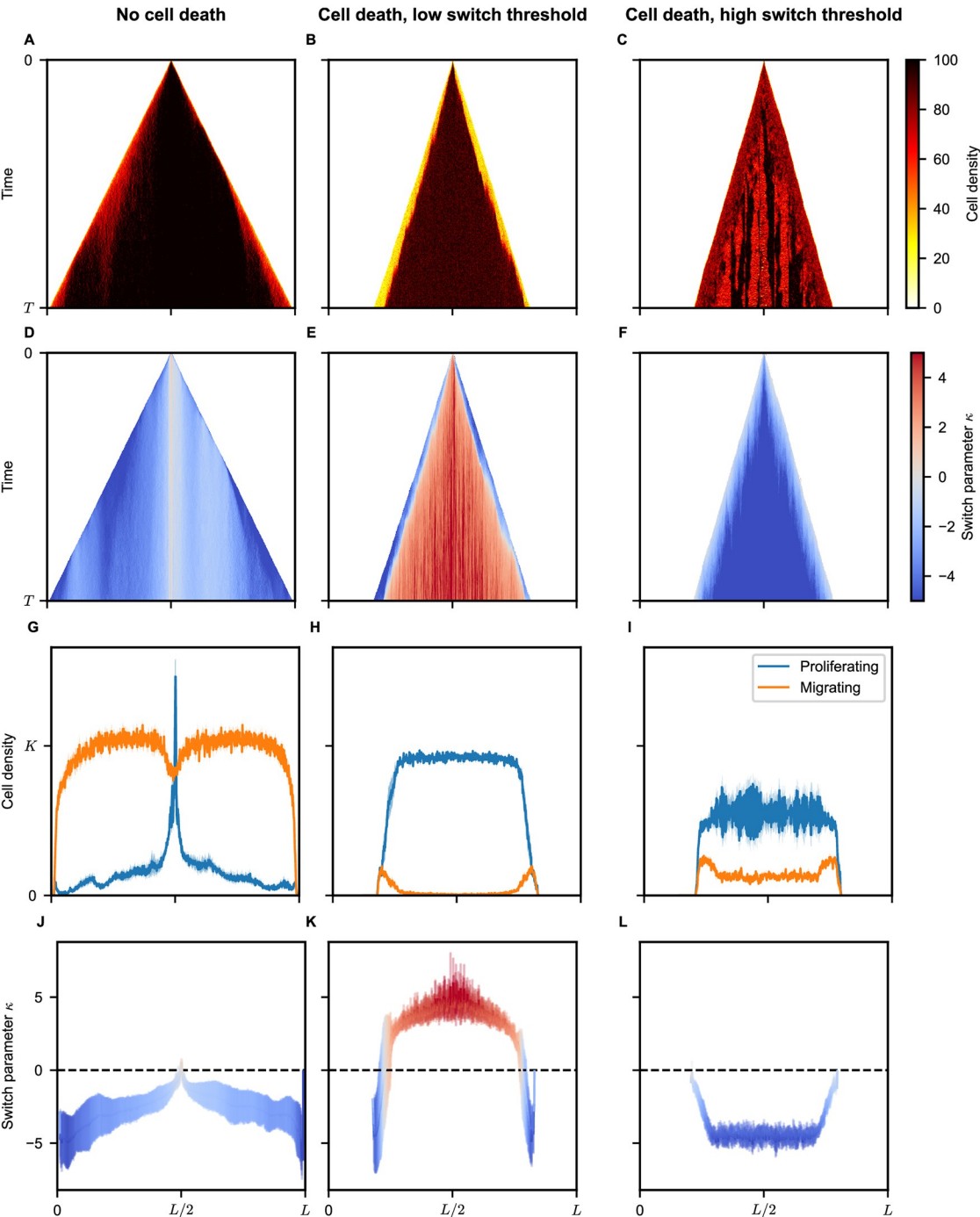

**Fig 3. Phenotypic and genetic heterogeneity emergence.** Example realizations (**A-F**) and ensemble averages (**G-L**) for three parameter sets. Evolution of cell density profiles (**A-C**) and local switch parameter (**D-F**). (**G-I**) Distribution of proliferating (blue) and migrating (orange) cells at the endpoint. (**J-L**) Switch parameter $\kappa$ distribution at the endpoint. Shaded regions represent one standard deviation. Negligible cell death (first column) results in attractive type cells ($\kappa > 0$) around the initial lattice node at $x = L/2$, with migrating phenotype cells outgrowing due to quick spread across the lattice (regime 1, Fig 4). High cell death and low switch threshold (second column) lead to predominantly proliferative phenotypes with attractive-type cells ($\kappa > 0$) dominating the tumor bulk and repulsive-type cells ($\kappa < 0$) invading the edge (regime 2, Fig 4). High cell death and high switch threshold (third column) result in a majority of proliferative cells but with a significant fraction of migrating cells throughout the tumor. Repulsive-type cells dominate the tumor, and weaker phenotype switch cells ($\kappa \approx 0$) are more likely at the tumor edge (regime 3, Fig 4).

the tumor bulk have a positive switch parameter $\kappa > 0$. Systematic variation of the threshold parameter $\theta$ and the death probability $\delta$ reveals distinct evolutionary regimes of the system.

**Negligible apoptosis rates favor the migratory phenotype.**   If there is negligible cell death ($\delta = 0$), the evolutionary pressure is low, and the relative cost of switching to the migratory phenotype is small. This situation could apply to the early stage of tumor growth, where resources are abundant, and there is only a minimal immune response. In this case, cells that spend more time migrating have an advantage because the best chance to outgrow competing cells is to migrate to empty space and start growing there. Therefore, in this regime, the fraction of cells in the migratory phenotype is the highest, see Figs 3G and S1B. For a small threshold parameter $\theta$, the evolutionary process favors cells of the repulsive type ($\kappa < 0$) because these cells start to migrate as soon as the local cell density exceeds the low threshold, see Fig 3D and 3J. For higher values of the threshold parameter $\theta$, the independent type ($\kappa \approx 0$) dominates, see S1D Fig and S2 Video. These cells switch between migrating and resting states irrespective of the local cell density. They have an advantage over the repulsive cells because the latter would only start moving at very high local cell densities $\rho_{\mathcal{N}} > \theta$ when the independent cells have already populated the local microenvironment. They also outgrow the attractive type ($\kappa > 0$) as these cells are predominantly in the migratory phenotypic state, can only grow in high-density environments, and can, therefore, not populate low-density areas.

**Low switch threshold favors repulsive strategies at the tumor edge and attractive strategies in the tumor bulk.**   The situation changes dramatically for higher values of the death rate and low switch threshold. This regime could apply to larger tumors with weak blood perfusion where nutrients and oxygen are scarce in the tumor bulk, see the discussion of clinical evidence below. The migratory phenotype now offers an advantage only if the cell is located near the tumor boundary so that low-density/high-oxygen areas are easily reachable, see Figs 3H and S2B and S2C. Therefore, cells adapt to their local environment: On the one hand, within the tumor, where the local cell density is high, cells evolve to favor the attractive strategy ($\kappa > 0$). As cell density is high and cell proliferation is low due to space restrictions, even if many cells switch to the proliferative phenotype, this resembles a dormant state, where cells wait for neighboring cells to die and to fill new gaps quickly. On the other hand, cells at the edge of the tumor evolve towards the repulsive genotype ($\kappa < 0$), which favors the migratory phenotype when the local cell density becomes too large, see Figs 3E, 3K, S2B and S2D, and S4 Video. Therefore, these cells can consistently grow at the edge of the tumor, although they would likely die out in the high-density tumor bulk on the search for a low-density environment.

**High switch threshold leads to repulsive strategies throughout the tumor.**   For high values of the switch threshold $\theta$, the situation resembles a well-perfused tumor, and the dynamics change again (see Clinical Evidence Section). In this case, the repulsive go-or-grow strategy ($\kappa < 0$) has a clear evolutionary advantage over all other strategies and is adopted throughout the tumor, see Figs 3F, 3L and S3D, and S6 Video. Cells with this strategy focus on growth while the local cell density is lower than the switch threshold $\theta$ and only switch to the migratory phenotype when the cell density becomes very high. However, significant heterogeneity persists in this regime; specifically, cells at the tumor periphery tend to exhibit elevated $\kappa$ values, predisposing them to a migratory phenotypic state more frequently than cells residing in the central mass of the tumor, see Figs 3I, S3B and S3D.

## A mean-field approximation can predict the evolutionary regimes

We derive a mean-field description of our model to elucidate the role and interactions between the different involved scales (genotype, phenotype, and microenvironment) following the

procedure in [23]. Here, mean-field refers to an approximation, where the expected value of a function $f$ of any stochastic variable $X$ is replaced by the function evaluated at the expected value of the stochastic variable, i. e., $\langle f(x) \rangle \approx f(\langle x \rangle)$. Using this approximation, we can derive the effective per-capita growth rate $F_\kappa(\rho, \rho_\mathcal{N})$ of cells of genotype $\kappa$ at a lattice node with cell density $\rho$ surrounded by a microenvironment of average cell density $\rho_\mathcal{N}$

$$F_\kappa(\rho, \rho_\mathcal{N}) = \alpha(1 - \rho)r_\kappa(\rho_\mathcal{N}) - \delta. \tag{5}$$

See S1 Text for a derivation. We interpret this function as a context-dependent fitness function, i. e., a fitness function which depends not only on a cell's genotype $\kappa$ but also on its microenvironment $\rho, \rho_\mathcal{N}$. To predict the outcome of the evolutionary dynamics, we look for its maxima in dependence on the local cell density and the switch threshold. The effective growth rate is maximal if cells spend most of the time in the proliferative state, i. e., $r_\kappa(\rho_\mathcal{N}) \to 1$.

$$r_\kappa(\rho_\mathcal{N}) \to 1 \begin{cases} \text{for} & \kappa \to \infty & \text{if} & \rho_\mathcal{N} - \theta > 0 \\ \text{for} & \kappa \to -\infty & \text{if} & \rho_\mathcal{N} - \theta < 0. \end{cases} \tag{6}$$

For the tumor bulk, we can estimate an upper limit for the average cell density in the neighborhood. To do this, we assume that the densities in the node of interest and its neighbors are at the maximum equilibrium density. The maximal equilibrium density can be reached if all cells in the respective nodes have the proliferative phenotype ($r_\kappa = 1$). Assuming this extreme case, we end up with the following upper limit to the local density

$$F_\kappa(\rho, \rho_\mathcal{N}) = 0 \Rightarrow \rho_\mathcal{N} \lesssim 1 - \frac{\delta}{\alpha} =: \rho_{\max}. \tag{7}$$

By setting $\rho_\mathcal{N} = \rho_{\max}$ in Eq 6, we can define two distinct regimes of the evolutionary dynamics in the tumor bulk, see black dashed line in Fig 4A. In the heatmap plot, we show the mean switch parameter $\kappa$ of all cells in the tumor at the end of the simulation for each parameter combination $(\theta, \delta)$. For $\rho_{\max} = 1 - \delta/\alpha > \theta$, we predict the evolutionary dynamics to drive the cells toward the attractive strategy ($\kappa > 0$, region 2 in Fig 4). Conversely, for high $\theta$ and high $\delta$, we predict that the repulsive strategy ($\kappa < 0$) is favored (region 3 in Fig 4). Finally, the situation without cell death ($\delta = 0$) represents a special case, see region 1 in Fig 4. Since there is no cell death, the evolutionary process is not determined by the slow dynamics in the tumor bulk. Instead, the tumor cells invade the surrounding tissue like a traveling wave, and the cells that do this the fastest initially will also dominate in the long term. Due to the phenotypic switch, we can describe the movement of tumor cells by a cell density-dependent diffusion constant and proliferation rate [23]. However, at the tumor front, we can make a low-density approximation, neglecting the cell density-dependence of proliferation and migration, i. e., $r_\kappa(\rho_\mathcal{N}) \approx r_\kappa(0)$. In this case, we obtain an effective proliferation rate proportional to the probability of being in the proliferative phenotype $\alpha_0 \propto r_\kappa(0)$. The diffusion constant, on the other hand, is proportional to the probability of being in the migratory phenotype $D_0 \propto (1 - r_\kappa(0))$. With this approximately constant proliferation rate and diffusion constant, we recover the classical Fisher-KPP equation of population growth in a homogeneous environment [34]. This traveling wave has a minimum wave speed that is proportional to the square root of the product of the diffusion constant and the proliferation rate, i. e.,

$$c_\kappa \propto \sqrt{\alpha_0 D_0} = \sqrt{r_\kappa(0)(1 - r_\kappa(0))}. \tag{8}$$

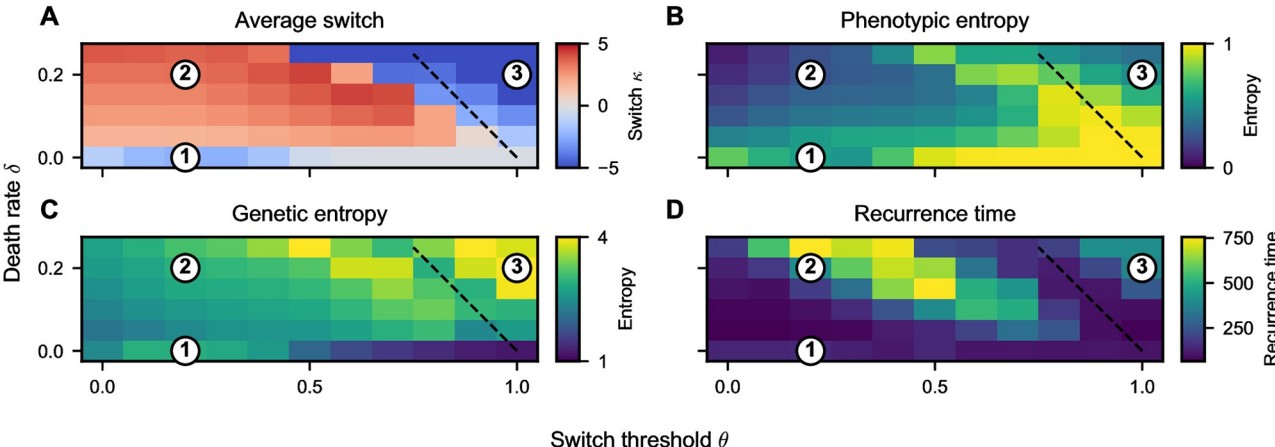

**Fig 4. Emerging heterogeneity and treatment responses depend on death rate and switch threshold as predicted by mean-field theory. (A)**
Average switch parameter $\kappa$ in the cell population. We can distinguish three different regimes. (1) If there is no cell death, the population is dominated by cells that can invade the surrounding tissue the fastest, which are cells with a switch parameter at zero or slightly below. (2) For a low switch threshold and non-zero death rate, the tumor is dominated by cells with an attractive go-or-grow strategy ($\kappa > 0$). However, there is considerable genetic heterogeneity, with repulsive cells at the tumor front, see Fig 3, second column. (3) With increasing switch threshold $\theta$ and death rate $\delta$, there is a transition to the repulsive regime, where all cells in the tumor use the repulsive strategy ($\kappa < 0$). This transition can be predicted by mean-field theory (black dashed line, given by Eqs 6 and 7. (B) The phenotypic heterogeneity is largest near the transition line between evolutionary regimes. (C) The genetic heterogeneity shows a bimodal behavior, with a minimum at the transition line surrounded by local maxima. (D) The recurrence time is longest in regime (2) and minimal near the transition line as well as for low values of the switch threshold $\theta$.

We are now looking for the switch parameter $\kappa$, which maximizes this wave speed, as the corresponding cancer cell population populates the tumor microenvironment fastest and dominates the tumor for the rest of its existence if there is no cell death. By calculating the maximum of Eq 8 we find that

$$c_\kappa \to \max \Leftrightarrow r_\kappa(0) = 1/2 \Rightarrow \kappa = 0 \tag{9}$$

maximizes the wave speed. This prediction agrees relatively well with numerical simulations, see Fig 4, regime 1. However, we observe a bias towards the repulsive strategy in numerical simulations. Describing this effect requires a more sophisticated mathematical analysis accounting for the density-dependence of the phenotypic switch, which is beyond the scope of this work.

## Phenotypic and genetic heterogeneity are predictors of treatment success

Next, we investigate the implications of the observed emergent heterogeneity on cancer treatment prospects. To do this, we simulate a simplified treatment scenario, which we apply to the synthetic tumors, see Methods. In short, starting from the simulation end points of the tumor growth simulations, we remove a large portion of tumor cells and record the recurrence dynamics. Most importantly, we record the recurrence time, which we define as the time until the tumor reaches the same cell number as before the treatment. The recurrence time shows considerable variability, ranging from below 100 time steps to more than 500 time steps or even complete tumor extinction. This corresponds to recurrence times of around one hundred days to more than two years, see S1 Text, which agrees well with empirical data on tumor progression of high-grade glioma [35]. We removed tumor extinction cases when performing the recurrence time statistical analysis. The recurrence time depends on the input parameters in a complex non-linear fashion, but we distinguish the following qualitative trends (Fig 4D):

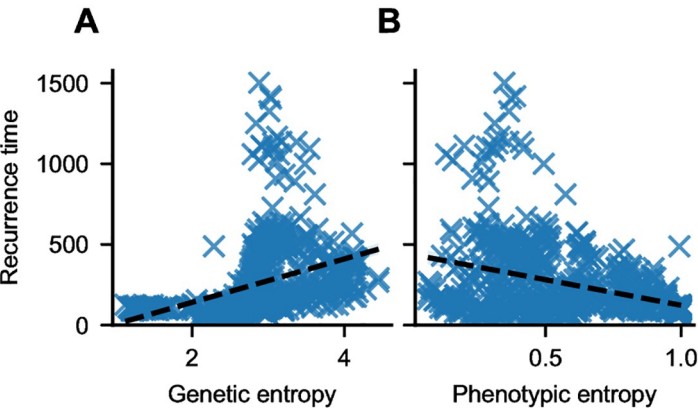

**Fig 5. Genetic and phenotypic heterogeneity predict treatment success.** Shown is the time to recurrence of synthetic tumors versus the genetic entropy (**A**) and phenotypic entropy (**B**). Higher phenotypic entropy is associated with lower recurrence times, while higher genetic entropy is associated with higher recurrence time, corresponding to better prognosis. Black dashed lines are linear regressions.

Recurrence is fastest for very low death rates and very low or very high switch thresholds as well as near the transition line between evolutionary regimes (black dotted line in Fig 4D). It is longest, even allowing for total tumor extinction in rare cases ($n = 4$ out of 660 simulations), in the attractive regime (2) in Fig 4, because these cells start migrating at low densities instead of growing. The tumor extinction is a consequence of the strong Allee effect affecting cells with the attractive go-or-grow strategy, as predicted previously [23]. Finally, the repulsive regime (3) in Fig 4 shows a medium recurrence time but no extinction.

So far, we described our results in dependence on model parameters, specifically the death rate and the density threshold. However, these parameters cannot be measured in a clinical setting. Therefore, we also quantify our synthetic tumors' phenotypic and genetic heterogeneity, as described in Methods. These observables can be recorded in a clinical setting and potentially hold prognostic value [33]. We find that the phenotypic heterogeneity, estimated as the entropy of the distribution of migrating and resting cells, was maximal near the transition line of evolutionary regimes and for very low death rates (Fig 4B). Conversely, the genetic entropy has a local minimum near the transition between evolutionary regimes (Fig 4C).

To mimic a medical scenario, we now treat our synthetic tumors as black boxes, i. e., we pretend that we do not know any cell-specific parameters. Instead, we assume that we can only measure the phenotypic and genetic heterogeneity and ask whether these observables hold prognostic value for the therapy outcome. Thus, we plot the recurrence time depending on genetic and phenotypic entropy (Fig 5). Interestingly, we find that a higher genetic entropy is associated with a longer recurrence time, i. e., a better prognosis (Fig 5A). On the other hand, high phenotypic entropy is correlated with poor treatment outcome, i. e., short recurrence times (Fig 5B).

## Clinical evidence of evolutionary regimes

As mentioned, the go-or-grow dichotomy is particularly relevant for glioblastoma tumors. Therefore, we turn to medical data of glioblastoma patients to collect evidence of distinct evolutionary regimes. One prognostic marker for glioblastoma patients is the *relative cerebral blood volume* (rCBV), which can be measured using magnetic resonance imaging (MRI). The rCBV indicates the blood perfusion of the tumor relative to healthy tissue. Patients can be

associated with one of two groups, depending on the rCBV value: An increased rCBV ($> 1.75$ [36]) is associated with a more aggressive tumor [37] with a worse prognosis and a short recurrence time after treatment [35]. Conversely, patients with low rCBV ($< 1.75$) have a better prognosis, i. e., longer recurrence times. In the context of our model, we argue that increased rCBV corresponds to a higher switch threshold $\theta$, i. e., increased cell density threshold, because nutrients and oxygen are more readily available in well-perfused tumors. In this case, tumor cells only experience a lack of nutrients when the cell density approaches the carrying capacity, i. e., $\rho \approx \theta \approx K$. Consequently, we associate high-rCBV tumors with the evolutionary regime (3), in Fig 4, i. e., evolving to a predominantly repulsive strategy with a shorter recurrence time. Patients with weak perfusion (rCBV $< 1.75$), on the other hand, are in regime (2), where most cells evolve to an attractive strategy, and only cells at the tumor periphery have the repulsive strategy.

Moreover, several studies have found that postoperative hypoxia and infarction (death of brain tissue due to insufficient blood supply after surgery) trigger invasive tumor growth correlated with multifocal and distant recurrence patterns, associated with shorter overall survival and post-progression survival, without a change of the recurrence time [38–40]. These findings agree with our model, assuming that postoperative hypoxia corresponds to decreased cell density threshold $\theta$. This, in turn, leads to increased cell migration of cells with the repulsive go-or-grow strategy, see Fig 2D, which are present in all tumors irrespective of the evolutionary regime, and can ultimately lead to multifocal/distant recurrence.

In summary, tumor recurrence patterns of glioblastoma patients hint at the existence of two evolutionary regimes defined by tumor perfusion, as predicted by our mathematical model.

## Discussion

Describing, reconstructing, or even predicting the evolutionary dynamics in tumors using mathematical models can potentially improve cancer therapy. However, this remains a formidable challenge due to the multiscale interplay of genetic, phenotypic, and microenvironmental heterogeneity. This study investigated the interplay of evolutionary dynamics and the phenotypic plasticity between migration and proliferation, which is especially relevant for glioblastoma cells. Specifically, we asked if an optimal regulation of the phenotypic switch exists and how it depends on environmental parameters. Moreover, we studied the clinical implications of this phenotypic plasticity. To this end, we applied a novel cellular automaton model for cells that switch between a migratory and proliferating phenotype depending on their microenvironment. Here, we used the local cell density as a proxy for other environmental factors. Cell density is compared to the switch threshold $\theta$. We enabled each cell to employ its unique go-or-grow strategy determined by the cell-specific phenotypic switch parameter $\kappa$, which we envision as a simplified representation of a cell's genotype. By doing so, we could distinguish three qualitatively different strategies: The independent strategy ($\kappa \approx 0$), where cells switch between the two phenotypes independent from the local microenvironment and both phenotypes are always equally likely; the attractive strategy ($\kappa > 0$), where cells are increasingly likely to be in the proliferative phenotype with increasing cell density; and finally, the repulsive strategy ($\kappa < 0$), with which cells are more and more likely to be in the migratory phenotype with increasing cell density. Both phenotypes have equal probability at the cell density threshold $\theta$, irrespective of $\kappa$. The switch parameter $\theta$ is fixed and assumed to be a function of the tumor microenvironment. Assuming that the phenotypic switch is triggered by low resource availability, we proposed that a large switch threshold $\kappa$ corresponds to a well-perfused tumor

in which nutrients and oxygen are readily available. Conversely, a low switch threshold indicates a low resource availability.

We then investigated the evolutionary dynamics in a growing tumor for varying death probability $\delta$ and switch threshold $\theta$ and observed the emergence of phenotypic and genetic heterogeneity. Notably, we found that for a low cell density threshold (resource scarcity), the attractive strategy is favored in the tumor bulk, while the repulsive strategy dominates at the tumor border. This situation is reversed in the case of a high switch threshold, which we could predict by a mean-field analysis. In both situations, cells are preferably in the proliferating phenotype in the tumor bulk and the migratory phenotype at the tumor border. However, the tumor's genetic makeup is fundamentally different in both cases and depends on the microenvironment, i. e., the apoptosis rate $\delta$ and the switch threshold $\theta$.

We also investigated a treatment scenario and recorded the response of synthetic tumors, specifically the recurrence time. We found that the recurrence time varies significantly, like for many cancers in medical practice [35, 41, 42]. Interestingly, high phenotypic heterogeneity was associated with a poor clinical outcome, while moderate genetic heterogeneity is correlated with better therapy success.

The evolution of phenotypic heterogeneity with more migratory cells at the tumor front and cells with higher proliferation rate in the tumor bulk had been observed in several previous modeling studies [20, 25, 26, 43]. However, so far, theoretical studies typically either study the effects of phenotypic plasticity or the evolution of genetic traits. They do not consider the cellular phenotype as the consequence of a cell decision-making process involving both the genotype and the cellular microenvironment. Here, we filled this gap by implicitly modeling the cell decision-making process in the context of the so-called go-or-grow dichotomy using a phenotypic switch function depending on cell density. This implies that we do not distinguish cancer cell clones that are inherently more migratory or have a higher proliferation rate. Instead, each cell switches between these two phenotypes as a response to its microenvironment and according to its genotype.

Experimentally identifying the phenotypic switch function would require studying the respective tumor cell lines in varying microenvironmental contexts.

While we did not focus on specific tumor cell lines here, it has been hypothesized before that the attractive strategy ($\kappa > 0$) corresponds to a low-grade tumor, and the repulsive strategy ($\kappa < 0$) to a high-grade tumor [23]. This aligns with clinical evidence on the effect of blood perfusion and postoperative infarction in glioblastoma tumors.

Our model also predicted that increased phenotypic heterogeneity correlates with a worse prognosis, in contrast to high genetic heterogeneity that can be associated with longer recurrence times. This agrees with results by Sharma et al. [44], who found that in non-small-cell lung cancer, genetic heterogeneity alone was insufficient to capture the heterogeneity observed at the transcriptomic level. They showed that phenotypic heterogeneity results from various intra- and extracellular sources. For example, the proliferation rates of cells of the same subclone varied depending on the tumor microenvironment. They concluded that tumor heterogeneity should be assessed at multiple levels to find the sources of phenotypic heterogeneity affecting clinical outcomes.

In our proposed model, we integrate the impact of the microenvironment on the phenotypic transition between motile and proliferative cellular behaviors through a dependency on local cell density. This assumption is reasonable, as other potential environmental factors, such as nutrient and oxygen availability, molecular signaling gradients, or additional cell-cell interactions, are also mediated by and correlated with local cell density. Simplifying the inherent complexity allows a more tractable exploration of fundamental organizational principles. We anticipate that significant population characteristics, such as the emergence of phenotypic and

genetic spatial heterogeneity, are still observable even if the reliance of plasticity on local density is refined into a more accurate dependency on specific cell-microenvironment interactions. Future research can build on this foundation by exploring additional aspects of tumor evolution and therapy response. For instance, incorporating explicit dependencies on specific cell-microenvironment interactions and additional environmental factors, such as diffusing nutrients and oxygen, could provide a more comprehensive understanding of tumor growth's complex dynamics. This would lead to a more complex, nonlinear, non-monotonic, phenotypic switch function with multiple inputs, potentially resulting in even richer dynamics resulting from the dynamic and heterogeneous tumor microenvironment.

In conclusion, we showed that the interplay of phenotypic plasticity and Darwinian evolution leads to multi-scale heterogeneity, i. e., heterogeneous spatial distributions of cancer genotypes and phenotypes, affecting cancer treatment outcomes. The therapy response of synthetic tumors showed considerable variability and depended on biological parameters in a complex, non-linear manner, reminiscent of clinical practice. Associating the recurrence time with genetic and phenotypic heterogeneity revealed that phenotypic heterogeneity is the more important prognostic marker associated with worse treatment outcomes.

## Supporting information

**S1 Text. Supplementary information.** The supplementary text contains 1) a list of mathematical symbols; 2) a table of the parameter values used; 3) a detailed mathematical LGCA model description; 4) a derivation of the average per-capita growth rate, i. e., the fitness function; 5) an estimation of the physical units of the LGCA time step length and lattice spacing.
(PDF)

**S1 Fig. 2-D growth dynamics without cell death.** Snapshot of an exemplary simulation on a hexagonal lattice without cell death corresponding to regime 1 in Fig 4. (**A**) Total cell density, (**B**) migratory cells, (**C**) proliferating cells and (**D**) average local switch parameter $\kappa$. Cells with the independent strategy ($\kappa \approx 0$) grow the fastest and dominate the tumor, leading to a mix of migratory and proliferative cells throughout the tumor. Parameters: $k = 300$, $K = 50$, $L = 250$, $\theta = 0.5$, $\delta = 0$.
(PNG)

**S2 Fig. 2-D dynamics with cell death and low phenotypic switch threshold.** Snapshot of an exemplary simulation on a hexagonal lattice with cell death and low phenotypic switch threshold corresponding to regime 2 in Fig 4. (**A**) Total cell density, (**B**) migratory cells, (**C**) proliferating cells and (**D**) average local switch parameter $\kappa$. Spatial heterogeneity of genotypes ($\kappa$ values), phenotypes, and density emerges, with a high-density tumor core of proliferating cells with an attractive strategy ($\kappa > 0$) surrounded by a low-density tumor rim of migratory cells with a repulsive strategy ($\kappa < 0$). Parameters: $k = 300$, $K = 50$, $L = 250$, $\theta = 0.2$, $\delta = 0.2$.
(PNG)

**S3 Fig. 2-D dynamics with cell death and high phenotypic switch threshold.** Snapshot of an exemplary simulation on a hexagonal lattice with cell death and low phenotypic switch threshold correponsing to regime 3 in Fig 4. (**A**) Total cell density, (**B**) migratory cells, (**C**) proliferating cells and (**D**) average local switch parameter $\kappa$. The whole tumor evolves towards the repulsive strategy, resulting in a homogeneous tumor of high density with migratory and proliferating cells throughout the tumor. Parameters: $k = 300$, $K = 50$, $L = 250$, $\theta = 0.9$, $\delta = 0.2$.
(PNG)

**S1 Video. Cell density in 2-D growth dynamics without cell death.**
(MP4)

**S2 Video. Distribution of switch threshold in 2-D growth dynamics without cell death.**
(MP4)

**S3 Video. Cell density in 2-D growth dynamics with cell death and low switch threshold.**
(MP4)

**S4 Video. Distribution of switch threshold in 2-D growth dynamics with cell death and low switch threshold.**
(MP4)

**S5 Video. Cell density in 2-D growth dynamics with cell death and high switch threshold.**
(MP4)

**S6 Video. Distribution of switch threshold in 2-D growth dynamics with cell death and high switch threshold.**
(MP4)

## Author Contributions

**Conceptualization:** Simon Syga, Haralampos Hatzikirou, Andreas Deutsch.

**Data curation:** Simon Syga.

**Formal analysis:** Simon Syga.

**Investigation:** Simon Syga, Harish P. Jain, Marcus Krellner.

**Methodology:** Simon Syga, Haralampos Hatzikirou.

**Project administration:** Andreas Deutsch.

**Software:** Simon Syga, Harish P. Jain.

**Supervision:** Andreas Deutsch.

**Visualization:** Simon Syga.

**Writing – original draft:** Simon Syga.

**Writing – review & editing:** Simon Syga, Harish P. Jain, Marcus Krellner, Haralampos Hatzikirou, Andreas Deutsch.

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
