## [Decision Letter · Decision Letter 0]

7 May 2024

Dear Mr Syga,

Thank you very much for submitting your manuscript "Evolution of phenotypic plasticity leads to tumor heterogeneity with implications for therapy" for consideration at PLOS Computational Biology. As with all papers reviewed by the journal, your manuscript was reviewed by members of the editorial board and by several independent reviewers. The reviewers appreciated the attention to an important topic. Based on the reviews, we are likely to accept this manuscript for publication, providing that you modify the manuscript according to the review recommendations.

Dear authors,

We have received the reviews we were expecting to decide about the publication your manuscript. As you see from their evaluations, we think that your article deserves publication under a minor/major revision, following the advices of the three reviewers.

Best regards,

Jean Clairambault, guest editor

Sincerely,

Jean Clairambault

Guest Editor

PLOS Computational Biology

Denise Kühnert

Section Editor

PLOS Computational Biology

Dear authors,

We have received the reviews we were expecting to decide about the publication your manuscript. As you see from their evaluations, we think that your article deserves publication under a minor/major revision, following the advices of the three reviewers.

Best regards,

Jean Clairambault, guest editor

Reviewer's Responses to Questions

**Comments to the Authors:**

Reviewer #1: Review-PLOS-Syga

The paper by Syga et al considers a lattice gas cellular automaton model for the phenotypic switching of cancer cells between a proliferative state and a state of propagation. Models of this type are known as go-or-grow models, and they have been used in the modelling of cancer for several years. In this paper the authors focus on the spatial heterogeneity that arises as result of phenotypic and genotypic switches, depending on the microenvironment. The paper is very well written and it shows some interesting results. I recommend publication after my comments below have been addressed.

My main comment relates to the use of kappa, the parameter in the switching probability function. I think its use should be better explained.

I was first confused as I realized in the results section that kappa changes over time. In fact, its the main point of the paper. It would be nice to mention this earlier, for example in line 107-109 you could add a comment saying that the changes of kappa during the experiments are the main focus here, or something like that.

Then I tried to understand how kappa changes, and it seems it only changes during a mitosis event, where a daughter cell randomly chooses a kappa “close” to the mother’s kappa. This gave me the following understanding. A mother that prefers to move, makes a quick stop to divide, and the daughter cells chose also a kappa that preferentially leads to movement. That’s why those kappa values accumulate at the tumor boundary? Is this the right understanding? In other words, from the explanations given I was not sure I understand te selection mechanisms for positive or negative kappa.

In the discussion you relate this process to cells decision making. I am not clear what you mean. Based on the above the cells make no decisions whatsoever. Once a kappa is chosen for a cell, it keeps it forever and doesn’t make any choice. The choice of go or grow is then given by a fixed probability. No more choice here. The kappa value is given to a cell on its birth, so there is no choice either. I think relating your work to cell decisions is overselling your results, as cell decisions are never modeled here.

Further comments

The model was already published in [23] and some analysis was done there. What is the new contribution here? Please make this more clear.

In this paper you do not mention if your model has an Allee effect or not. But the same model in [23] has an Allee effect, hence it should play a role here too? Or not? Please explain.

In equation (2) you only consider monotonic probabilities. Have you thought about non-monotonic functions? Maybe that’s a way to model an optimal cell density, that the cells prefer to have.

Missing closing bracket in line 267

Mean field approximation: While reading this part I was expecting to see a dynamic equation for kappa, like kappa_t = …. Is this possible? Could be interesting.

Simulations: Have you done 2-D simulations? It would be nice to see the expanding ring with propagating cells on the outside rim and proliferating cells in the inside. If you have 2-D sims I recommend to add them.

Discussion: Please revise the statements on “cell decision making”

Discussion: The discussion is quite repetitive. For example lines 412-416, and 443-447 were already mentioned in the discussion earlier.

Supplement: The supplement is quite useful, so I had a look as well and it is fine.

Reviewer #2: This paper presents a detailed investigation into the evolutionary dynamics of tumor growth, focusing on glioblastoma tumors. The authors employ a mathematical modeling approach to explore the interplay between genetic and phenotypic heterogeneity, and the go-or-grow dichotomy in relation to microenvironmental influences.

The topic of the paper is interesting and addresses a challenging issue in mathematical modeling of cancer and clinical research. The paper is persuasive and clear for the most part. The study is well-structured and considers different scenarios, varying a switching parameter related to different possible tumor microenvironment. Overall, the manuscript presents interesting insights into tumor evolution and its clinical implications. The study incorporates both theoretical modeling with numerical simulations and experimental observations, enhancing the robustness of the findings.

However, some minor revisions should be addressed before publication.

- Although the paper deals with a qualitative rather than a quantitative model, it would benefit from discussing the estimation of model parameters through experimental settings and further validation against clinical data or experimental results. Particularly, it should be discussed whether the values of parameters chosen in Table S2 in the Supplementary Information are model-specific or chosen within a reasonable biological range. If chosen within a biological range, the biological references for these parameters should be added. Moreover, the units of measure should be included in the table.

- It would be interesting to report the computational cost of 1D simulations and discuss the potential benefits of extending the model to a 2D setting. Commenting on the computational challenges and advantages of transitioning to a higher-dimensional space, as well as discussing how a 2D model could capture spatial complexities and interactions more realistically, would be relevant.

- Including perspectives on future research directions would strengthen the conclusion of the paper. For instance, in the current work, the authors assume a constant environment and thus a constant switching function. However, there is growing evidence of the critical role of a heterogeneous and dynamic environment, such as in relation to oxygen distribution and hypoxia or glucose and nutrients. Discussing on this point and exploring how the model could be adapted to incorporate environmental heterogeneity and how this might influence tumor dynamics would be insightful.

In conclusion, while the paper is an interesting piece of research, addressing these suggestions will strengthen the paper's impact and relevance to the field. Therefore, I recommend the manuscript for publication after addressing these minor revisions.

Reviewer #3: This paper deals with an essential, and yet unsolved, problem in oncology, namely the effect of heterogeneity in the growth and response to therapy of tumours. This paper presents a relatively simple model of tumour growth. The simplifications introduced allow for a degree of analytical progress which helps to understand and interpret the simulation results. This paper is worthy contribution to the existing literature in the subject.

I have only one minor comment regarding the model formulation that needs some clarification. It refers to the hypothesis that local density is a proxy for the effect of the microenvironment. This assumption is a reasonable one and allows the author to gain an analytical handle on the model. However, the relation between cell behaviour and the microenvironment is a complex, non-linear one. It would be useful to include a discussion on the limitations of this asumption and how the model could be modified to account for more detailed mathematical descriptions of the cell-to-environment relationship.

**Have the authors made all data and (if applicable) computational code underlying the findings in their manuscript fully available?**

Reviewer #1: **No: **no code provided

Reviewer #2: Yes

Reviewer #3: Yes

PLOS authors have the option to publish the peer review history of their article (what does this mean?). If published, this will include your full peer review and any attached files.

Reviewer #1: **Yes: **Thomas Hillen

Reviewer #2: **Yes: **Marcello Delitala

Reviewer #3: No

Figure Files:

Data Requirements:

Reproducibility:

References:

---

## [Decision Letter · Decision Letter 1]

23 Jul 2024

Dear Mr Syga,

We are pleased to inform you that your manuscript 'Evolution of phenotypic plasticity leads to tumor heterogeneity with implications for therapy' has been provisionally accepted for publication in PLOS Computational Biology.

Best regards,

Jean Clairambault

Guest Editor

PLOS Computational Biology

Denise Kühnert

Section Editor

PLOS Computational Biology

Reviewer's Responses to Questions

**Comments to the Authors:**

Reviewer #1: Thank you for the detailed changes. The paper can be accepted

Reviewer #2: All the points I have raised have been clarified and I have no further comments. The paper in the current, in my opinion, it is worth to be published in PLOS Computational Biology.

**Have the authors made all data and (if applicable) computational code underlying the findings in their manuscript fully available?**

Reviewer #1: None

Reviewer #2: Yes

PLOS authors have the option to publish the peer review history of their article (what does this mean?). If published, this will include your full peer review and any attached files.

Reviewer #1: **Yes: **Thomas Hillen

Reviewer #2: **Yes: **Marcello Delitala

---

## [Editor Report · Acceptance letter]

5 Aug 2024

PCOMPBIOL-D-24-00459R1 

Evolution of phenotypic plasticity leads to tumor heterogeneity with implications for therapy

Dear Dr Syga,

I am pleased to inform you that your manuscript has been formally accepted for publication in PLOS Computational Biology. Your manuscript is now with our production department and you will be notified of the publication date in due course.

With kind regards,

Anita Estes
